**EMBO** *reports*

# Keeping up with the neighbours: local synchronisation of cell fate decisions during development

Sally Lowell [ID] [1,2] [✉]

## Abstract

**Even before the advent of multicellular life, unicellular creatures would communicate with their neighbours to coordinate their behaviours. Multicellular organisms have the particular challenge of orchestrating the differentiation of stem and progenitor cells to generate and maintain coherent functional tissues. However, stem and progenitor cells face a problem: their differentiation response can be buffeted by oscillations or stochastic fluctuations in intrinsic regulators. This generates cell-to-cell variability, which can be further compounded when extrinsic cues don't provide clear unambiguous instructions. So, left to their own devices, cells may differentiate at different rates or different directions even in response to the same cues. Fortunately, cells in multicellular organisms are not left to their own devices: they continually sense and respond to the behaviours of their neighbours. Here I discuss when, where, and how stem and progenitor cells communicate to synchronise their response to differentiation cues. I highlight technical challenges in identifying such synchronisation mechanisms, and survey emerging technologies that may help overcome these challenges.**

**Keywords** Development; Differentiation; Stem Cells; Patterning; Synchronisation
**Subject Categories** Development; Signal Transduction; Stem Cells & Regenerative Medicine

## The need to synchronise differentiation

### Natural cell-cell variability in the timing of differentiation

It seems self-evident that all cell state transitions will exhibit some degree of cell-to-cell variability due to the stochastic nature of biochemical interactions (Huang, 2009) and the influences of oscillatory processes such as the cell cycle (Hardwick et al, 2018; Zinner et al, 2020). Stochastic variability in the differentiation response becomes more likely in situations where extrinsic differentiation cues fluctuate or conflict, for example, at threshold concentrations of morphogens (Kerszberg and Wolpert, 2007; Lander, 2013; Briscoe and Small, 2015; Lee et al, 2021; Kramer et al, 2022). Furthermore, progenitor cells sometimes migrate to new locations, and may need to adapt the pace or direction of differentiation to fit in with their new neighbours (Fulton et al, 2022; Lowell and Blin, 2022). This raises the question: in what situations is local variability in differentiation a problem that the embryo needs to solve?

## The pros and cons of synchronising differentiation

It is not always helpful to suppress cell-cell variability in differentiation. For example, if all cells in a population do the same thing at the same time, this may reduce the capacity for 'bet-hedging' and error-correction (Paszek et al, 2010; Levy, 2016). Indeed, recent data support the idea that temporal variability is desirable for robust development: experimentally enforcing synchronous behaviours during preimplantation mouse development has deleterious effects (Fabrèges et al, 2024).

Furthermore, progenitor cells sometimes need to generate more than one cell type in the same region. In these cases, variability can be deliberately amplified to generate patterned arrangements of distinct cell types. For example **lateral inhibition** mechanisms (see Box 1) ensure that some cells remain as progenitors while others differentiate (Mesa et al, 2018; Henrique and Schweisguth, 2019; Raina et al, 2021), while **Turing patterning** mechanisms (see Box 1) generate striped or patchy patterns of different cell types from initially homogenous populations (Marcon and Sharpe, 2012; Green and Sharpe, 2015). More generally, intrinsic variability enables symmetry breaking, allowing complexity to emerge even in the absence of pre-existing signalling centres (Serra et al, 2019; Journot et al, 2025; Schwayer et al, 2025).

In other cases, though, tissues emerge as coherent groups of cells of the same type (Gurdon et al, 1993). It therefore seems likely that cells, at least sometimes, need to communicate with each other to counteract variability in the pace of differentiation. In some cases, the job of orchestrating differentiation is assigned to special groups of 'influencer' cells. These form signalling centres which coordinate differentiation of the surrounding tissue, sometimes over long distances (Arnold and Robertson, 2009; Robb and Tam, 2004; Briscoe and Small, 2015). In other cases, groups of cells take equal

[1]Centre for Regenerative Medicine, Institute for Regeneration and Repair, The University of Edinburgh, Edinburgh EH16 4UU, UK. [2]Institute for Stem Cell Research, School of Biological Sciences, The University of Edinburgh, Edinburgh EH16 4UU, UK. [✉]E-mail: sally.lowell@ed.ac.uk

**Box 1 GLOSSARY**

**Community effect**: A process whereby a critical mass of similar progenitor cells needs to interact with each other in order to complete a differentiation process (Gurdon, 1988).

**Homoiogenetic induction**: A process whereby a relatively large number of differentiated cells induce nearby cells to differentiate into the same cell types as themselves (Gurdon et al, 1993).

**Lateral inhibition**: A process by which a differentiating cell can inhibit a neighbour from differentiating at the same time or in the same direction as itself (Henrique and Schweisguth, 2019).

**Lateral induction**: A process by which a differentiating cell encourages a neighbour to differentiate at the same time and in the same direction as itself (Daudet and Lewis, 2005).

**PUFFFIN** (Positive Ultrabright Fluorescent Fusion For Identifying Neighbours): a single-plasmid colour-of-choice neighbour labelling that is designed for use in developmental model systems.

**Quorum sensing**: A mechanism by which groups of cells adopt a particular behaviour (e.g. differentiation) only when surrounded by a critical mass of similar cells (Waters and Bassler, 2005).

**Synchronisation**: Here, we define local synchronisation as a process whereby cells differentiate at approximately the same time and in the same direction as each other.

**SynNotch** (Synthetic Notch): a synthetic sense-and-respond system that uses a re-engineered Notch receptor system to activate a transgene upon binding to a transmembrane synthetic ligand on 'sender' cells (Morsut et al, 2016). SynNotch has been adapted for use in developmental systems (He et al, 2017; Zhang et al, 2022; Malaguti et al, 2022).

**Turing patterning**: A process by which patterned differences between cells can emerge from initially homogenous groups of cells, based on feedback between molecules that diffuse between cells at different rates (Marcon and Sharpe, 2012).

responsibility for coordinating with each other, each cell sensing and responding to the state of its local neighbourhood (Fig. 1A–C) (Chubb et al, 2021). These categories are not mutually exclusive: for example, cells likely coordinate locally in order to synchronise their response to long-range graded signals (Lander, 2013). In this review, I focus on the question of how cells sense and respond to their neighbours in order to locally **synchronise** cell fate transitions (see Box 1).

## Examples from early vertebrate development

The mammalian post-implantation epiblast is an example of a tissue composed of a homogenous group of formative pluripotent cells that emerge relatively synchronously (Nichols and Smith, 2009; Smith, 2017; Morgani and Hadjantonakis, 2020). This tissue represents a pristine substrate for subsequent elaboration of the body plan, so it is critical that it is formed in a timely and coherent manner. It is built during implantation from a population of naive pluripotent cells, which rewire their intrinsic gene regulatory network to transition into a formative post-implantation pluripotent state (Smith, 2017). Measuring markers of naive and formative pluripotency reveals that this transition is initiated asynchronously but quickly becomes synchronised, with all cells completing the transition within a 24 h period in the mouse (Acampora et al, 2012) (Fig. 1D,E). This is further confirmed using rigorous clonal assays to examine changes in potency at regular intervals during implantation (Boroviak et al, 2014).

The synchrony of this transition may seem unsurprising, because all cells are likely exposed to similar environmental cues. It therefore came as a surprise to discover that this transition become desynchronised when naive pluripotent cells are separated from their neighbours and plated in sparse monolayer cultures: under these conditions, some cells differentiate within 12 h but some particularly slow-paced cells can take nearly 48 h to complete the transition, even though all cells start in the same state and are exposed to the same differentiation conditions (Betschinger et al, 2013; Kalkan et al, 2017; Mulas et al, 2017).

How do cells in the peri-implantation embryo overcome this intrinsic variability in the pace of differentiation? It seems that cells that differentiate too quickly during peri-implantation development can sense that they need to 'slow down' to restore an appropriate pace of differentiation. In this case 'sensing' mechanism is based on detecting levels of Nodal secreted by surrounding cells: cells upregulate the anti-differentiation factor Id1 if Nodal has not yet reached the levels required to support a post-implantation identity (Malaguti et al, 2019). Conversely, slow-differentiating naive pluripotent cells are sometimes able to sense that a neighbouring cell is differentiating more rapidly than themselves and respond by speeding up their own differentiation rate to 'catch up' (Fig. 1B) (Lebek et al, 2024).

Lineage commitment at gastrulation also seems to benefit from local synchronisation mechanisms. For example, pluripotent cells with mutations in Tet1, Tet2 and Tet3 (TKO cells) exhibit defects in both the speed and direction of differentiation (Pantier et al, 2024; Argelaguet et al, 2019). Careful scRNAseq analysis of chimeric embryos indicates that the usually slow-differentiating TKO mutant cells can restore a normal pace of differentiation when surrounded by wild-type cells (Cheng et al, 2022).

Gastrulation is characterised by the ingression of mesoderm-fated cells at the primitive streak. Live imaging of mesoderm ingression in chick embryos suggests that this process is initially a sporadic and reversible process, but gradually becomes locally synchronised, presumably as a consequence of local cell-cell communication. Grafting experiments confirm a positive feedback mechanism whereby early-ingressing mesoderm cells encourage mesoderm differentiation in their neighbours to coordinate the formation of the primitive streak (Voiculescu et al, 2014).

We still have only a limited understanding of the rules by which cells adjust the pace of differentiation to fit with their local neighbourhood. For example when is a local mis-match in the pace of differentiation corrected through local communication between cells (Lebek et al, 2024; Cheng et al, 2022; Voiculescu et al, 2014) and when does it trigger elimination of mis-differentiating cells by cell competition (Díaz-Díaz et al, 2017; Bowling et al, 2019; Nichols et al, 2022; Khandekar and Ellis, 2024)? Who synchronises with whom: i.e. when do faster cells slow down, and when do slower cells speed up? What are the cellular and molecular mechanisms that implement these rules?

## Strategies for synchronising differentiation

This section outlines various strategies that cells could use to synchronise differentiation. For convenience, these strategies are categorised here according to the terminology used by the authors

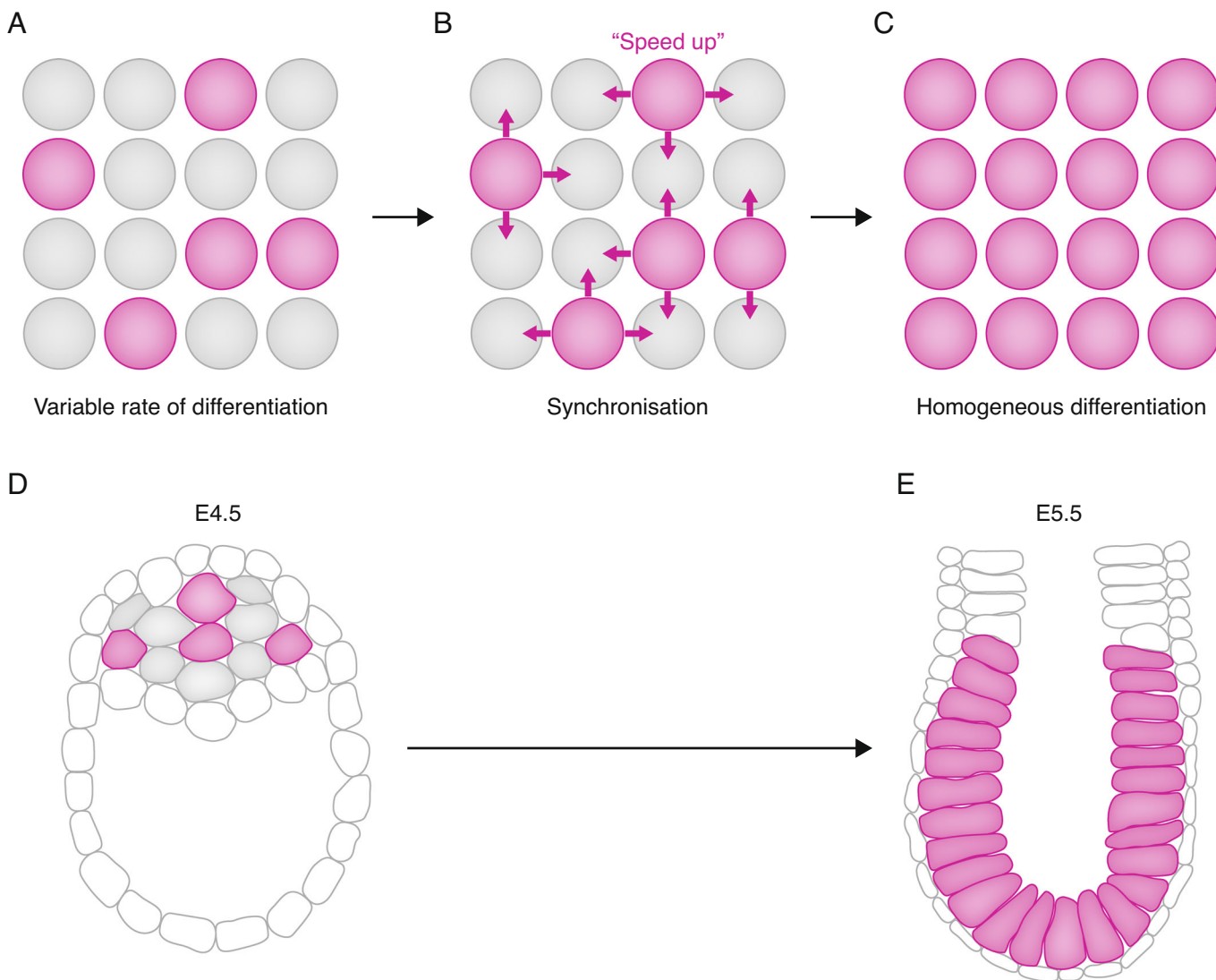

**Figure 1. Local synchronisation and the example of the epiblast.**

(A–C) Variability in the rate of differentiation (A) is counteracted by communication between cells (B) to achieve more homogenous and synchronous differentiation (C). (D) In the late preimplantation mouse epiblast (E4.5), only a subset of cells (purple) have exited naive pluripotency (referred to here as being differentiated). Note that this stage of development is after segregation of primitive endoderm, and that variability in cell state at this stage should not be confused with variability in the emergence of PE and EPI in the earlier blastocyst. (E) In the early post-implantation epiblast (E5.5) all cells have exited naive pluripotency (referred to hear as being differentiated). See the text for more details, and for evidence that this transition is actively synchronised cells sensing more-differentiated neighbours to synchronise differentiation throughout the epiblast. Grey: undifferentiated Purple: differentiated (in D, E, 'differentiated' has the meaning 'exited naive pluripotency').

who first reported these observations. However, in practice, these categories are not distinct and often share common features.

## Community effects

**Community effects** (see also Box 1) describe mutual-positive feedback mechanisms that result in decisive and uniform responses by a critical mass of similar cells (Fig. 2A). For example, using grafting experiments in Xenopus embryo, John Gurdon reported 'an effect in which the ability of a cell to respond to induction by differentiating as muscle is enhanced by, or even dependent on, other neighbouring cells differentiating' (Gurdon, 1988). Essentially, large grafts of animal cap tissue can change their identity more readily than smaller grafts can,

when either grafted to ectopic vegetal locations or 'sandwiched' between pieces of vegetal tissue (Fig. 2B).

Gurdon speculated that large groups of cells could perhaps accumulate high concentrations of a putative secreted pro-differentiation factor (Gurdon, 1988). He proposed that such 'community effects' may encourage decisive and coherent behaviours in situations where tissues form as homogenous blocks of cells (Gurdon, 1988; Gurdon et al, 1993).

Similar observations have been made in experiments using explants of paraxial mesoderm or limb bud from mouse embryos: mesoderm differentiation was only able to proceed if explants contained at least 30–40 cells (Cossu et al, 1995). In another example, rhombomeres form coherent groups of distinct identities,

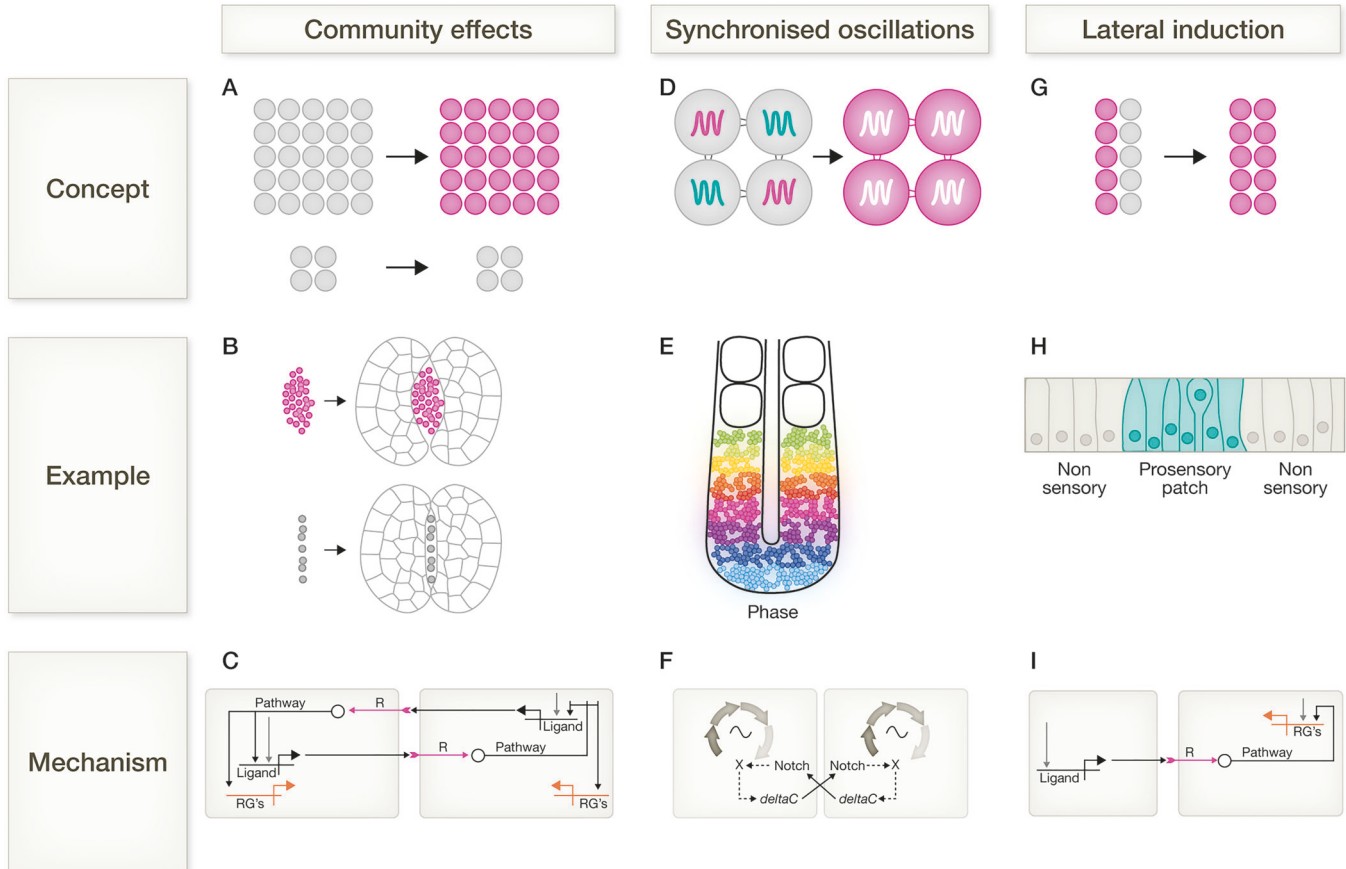

**Figure 2. Strategies for synchronising differentiation.**

(A) Community effects describe situations where large groups of like-cells differentiate efficiently but smaller groups do not. (B) Large groups of reaggregated *Xenopus* animal cap cells can change identity (indicated by a change of grey to purple) when sandwiched between explants of vegetal tissue, but small groups of the same cell type can not. Figure adapted from (Gurdon, 1988). (C) Mechanism proposed by (Bolouri and Davidson, 2010) to underlie a community effect. A pro-differentiation signalling ligand signals to neighbouring cells to activate expression of its own gene (and also to activate a differentiation programme). This results in mutual amplification of differentiation cues. Figure adapted from (Bolouri and Davidson, 2010). (D) Some progenitor cells exhibit intrinsic oscillations of differentiation regulators: when these oscillations are coupled between cells, this helps synchronise differentiation. (E) Oscillations of Hes1-Achilles reporter within the presomitic mesoderm of a mouse embryo. The phase of the oscillations are indicated by colour: cells in similar regions are in phase with each other. White squares indicate already-formed somites in the anterior region. Figure adapted from (Yoshioka-Kobayashi et al, 2020). (F) Mechanism proposed by (Jiang et al, 2000) to underlie synchronisation of the somitogenesis clock. Oscillatory expression of Notch target genes is synchronised between neighbouring cells via juxtacrine Notch signalling. Figure adapted from (Jiang et al, 2000). (G) Lateral induction describes a situation where differentiating cells induce differentiation in their neighbours. (H) Sensory cells in the inner ear of the chick form in coherent patches. Figure adapted from (Daudet and Lewis, 2005). (I) Mechanism underlying lateral inhibition. A differentiating cell upregulates a ligand that induces differentiation in neighbouring cells. Figure adapted from (Bolouri and Davidson, 2010).

in part because of a retinoic-acid-mediated community effect that encourages cells to adopt the same identity as their neighbours (Addison et al, 2018)

Overall, though, rather few examples of community effects have been reported in the decades since Gurdon's original paper (Guthrie et al, 1992; Couly et al, 1998; Trainor and Krumlauf, 2000; Busby et al, 2024), perhaps because many developmental systems are not readily amenable to the grafting approaches used by Gurdon. However, other creative experimental approaches can be used to identify community-effect-like behaviours: for example, micropatterning approaches were used to demonstrate that large colonies of human ES cells respond more robustly to differentiation cues than smaller colonies do (Nemashkalo et al, 2017).

What is the mechanistic basis for community effects? Various theoretical frameworks have been proposed (Bolouri and Davidson, 2010; Saka et al, 2011; Olimpio et al, 2018), centred on the idea that signalling pathways can upregulate expression of their own ligands such that cells cooperate to mutually amplify signalling (Fig. 2C). Such mutual-positive-feedback mechanisms may, for example, help amplify Nodal signalling during differentiation of the oral ectoderm in sea urchin (Bolouri and Davidson, 2010) or FGF signalling during differentiation of pluripotent cells (Gattiglio et al, 2023). This could, in principle, be extended to include any chemical or mechanical amplification mechanism that is sensitive to the number of participants (Raffaelli and Stern, 2020).

## Quorum sensing

**Quorum sensing** (see also Box 1) is a term predominantly used to describe the behaviour of unicellular organisms (Waters and

Bassler, 2005; Papenfort and Bassler, 2016; Gregor et al, 2010), but has also been applied to cells in multicellular organisms (see examples below). It describes a process by which cells can sense when they are surrounded by a large number of other individuals of the same type. These cells then respond by activating processes that are beneficial only when operated by a critical mass of individuals. In the case of unicellular organisms, these processes include bioluminescence, acquisition of virulence, and biofilm production. Quorum sensing isn't limited to the behaviours of individual cells: social insects use a form of quorum sensing to decide where to establish nests (Franks et al, 2006; Visscher, 2007). Mechanistically, quorum sensing operates by secretion of 'autoinducers', which only reach effective concentrations when relatively large numbers of individuals are secreting them (Waters and Bassler, 2005).

A number of phenomena described as 'quorum sensing' have been reported in multicellular organisms. For example, hair follicle regeneration in vertebrate skin depends on cooperation between nearby hair follicles using a quorum-sensing type mechanism: plucking large numbers of hairs triggers a disproportionally stronger regenerative response compared with picking smaller numbers of hairs. In this case, communication is mediated over relatively long distances, at least in part via the immune response. Hair-plucking triggers the release of CCL2, which can recruit macrophages if it accumulates to a sufficient level. These macrophages secrete the pro-regenerative signal TNFα to support hair regrowth across the local region. It has been proposed that this mechanism ensures a strong regenerative response to major injury while avoiding the cost of an unnecessarily strong response to smaller injuries (Chen et al, 2015).

In the immune system, T cells need to make coherent decisions about which type of effector cell to differentiate into, but face the problem of interpreting fluctuating or conflicting information. Mathematical modelling indicates that this could be explained by a quorum-sensing-like mechanism arising from feedback in signalling and transcriptional networks (Schrom et al, 2020). It seems likely that conceptually similar feedback systems may explain how progenitor cells during development coordinate differentiation decisions in the face of incomplete or conflicting inputs.

Spermatogonial stem cells also make use of a form of quorum sensing, in that they use competition for limited mitogens as a mechanism to detect the size of the stem cell compartment and adjust their behaviour accordingly (Kitadate et al, 2019). This serves the purpose of balancing self-renewal with differentiation rather than synchronising differentiation, but does serve as another example of how cells mutually orchestrate their decisions by quorum sensing rather than relying on an external spatially restricted instruction.

Finally, it has been proposed that differentiating embryonic stem cells exhibit something akin to a quorum-sensing mechanism. Mathematical modelling suggests that cells 'pool' information from autocrine FG4F in order to sense that they are part of a large group, and this might explain why large groups of cells survive and proliferate during differentiation while smaller groups of cells do not (Daneshpour et al, 2023). The mathematical model predicts that this information is shared across mm-scales (Daneshpour et al, 2023), yet other work demonstrates that FGF4 signals only to direct neighbours in cultures of embryonic stem cells (Raina et al, 2021). Can quorum sensing operate through cell-to-cell propagation of signalling (rather than diffusion of FGF4 across longer distances)?

Or does it depend on the build-up of FGF4 within the lumen of the embryo, or the culture medium of ES cells in vitro, as has recently been suggested (Schröter et al, 2023)?

Quorum sensing seems conceptually similar to a community effect: both encourage groups of cells to differentiate towards the same fate at the same time (Fig. 2A). One partial difference may be that quorum-sensing tends to describe a self-contained system based on local accumulation of threshold-dependent signal, while community effects can (sometimes) also describe changes in responsiveness to an extrinsic signal such as a morphogen. Nevertheless, there seems to be considerable overlap in the use of the two terms. Regardless of the particular terminology used, it seems likely that mutual sharing of pro-differentiation cues by a critical mass of cells may be an underappreciated mechanism for local coordination of cell fate decisions.

## Coupling oscillations

Some progenitor cells contain molecular oscillators: periodic increases and decreases in expression of particular genes, driven by negative feedback coupled to time delays in transcription and/or translation (Bosman and Sonnen, 2022; Chandel et al, 2024). Oscillations in genes encoding differentiation regulators can influence the probability of a cell differentiating at any given time. Cell-cell coupling of these oscillations therefore offers an opportunity to synchronise differentiation responses across a field of cells (Isomura and Kageyama, 2014; Bosman and Sonnen, 2022; Chandel et al, 2024) (Fig. 2D).

One striking example of this phenomenon is the vertebrate segmentation clock, which is characterised by oscillations in components and targets of the Notch, Wnt, and FGF signalling pathways (Venzin and Oates, 2020; Sonnen et al, 2018). These oscillations regulate the periodic anterior-to-posterior emergence of paired somites from the presomitic mesoderm of vertebrate embryos. Live imaging of sensitive fluorescent reporters of gene or protein expression can be used to visualise beautifully synchronised clock oscillations in vertebrate embryos or cell-based models of development (Yoshioka-Kobayashi and Kageyama, 2021) (Fig. 2E).

These oscillations are regulated at least partly cell-autonomously: they occur even in individual cells isolated from zebrafish presomitic mesoderm (Webb et al, 2016), or in cultures of presomitic mesoderm isolated from mouse embryos (Tsiairis and Aulehla, 2016), but tend to be uncoordinated when cultures are sparse (Webb et al, 2016). In contrast, in intact embryos, confluent monolayers, or 3D models of human development, oscillations become coordinated between neighbours to generate coherent waves of gene expression (Aulehla et al, 2008; Soroldoni et al, 2014; Yoshioka-Kobayashi et al, 2020; Matsuda et al, 2020; Diaz-Cuadros et al, 2020).

Local synchronisation of oscillations depends, at least in part, on Notch signalling (Jiang et al, 2000; Soza-Ried et al, 2014; Tsiairis and Aulehla, 2016; Webb et al, 2016; Hubaud et al, 2017) (Fig. 2F). It seems likely that this cell-cell coupling is critical for the coherent and timely emergence of somites. In keeping with this idea, experimental inhibition of Notch disrupts somitogenesis, although it should be noted that Notch likely plays multiple roles during somitogenesis, making Notch phenotypes complex to interpret (Venzin and Oates, 2020).

Differentiation of progenitors in the developing neural tube is governed by bHLH transcription factors, and several of these

factors exhibit oscillatory expression. These oscillations generally appear to be unsynchronised, not forming obvious coherent waves like those described above in the presomitic mesoderm (Imayoshi et al, 2013). For example, live imaging the mouse ventral neural tube, using Hes5-Venus reporter mice, reveals Hes5 oscillations that are not globally coordinated across the entire tissue. However, close inspection of individual cells over time reveals local 'microclusters, typically containing four to six cells, that oscillate in concert. These microclusters are organised periodically across the neural tube.

The spatial organisation of these locally synchronised microclusters depend in part on Notch activity (Hawley et al, 2025; Biga et al, 2021) and seems to be required for the spatiotemporal organisation of neuronal differentiation. Careful quantitative analysis and computational modelling suggest that local weak coupling of oscillations may be important for tuning the rate of differentiation as well as coordinating differentiation locally within the neural tube (Biga et al, 2021).

How many other developing tissues use coupled oscillations of cell fate regulators to coordinate differentiation? The findings described above, in the presomitic mesoderm and neural tube, were made possible thanks to sensitive live reporters using destabilised fluorescent proteins. These oscillations would not have been observed using standard fluorescent proteins, which are too stable to report on changes on relatively short time scales. Where oscillations were not obviously synchronised across the tissue, there is also the daunting technical challenge of quantifying the dynamic behaviours of individual cells within dense tissues and then interpreting the highly complex data that emerged from these analyses (Biga et al, 2021). These various technical challenges mean it is far from straightforward to detect oscillators in other tissues, although other examples are emerging, for example, in the intestine (Weterings et al, 2024), pancreas (Seymour et al, 2020), and muscle (Lahmann et al, 2019; Zhang et al, 2021). It will be interesting, in future work, to assess if, when, and where these oscillators become locally synchronised, and to investigate how these processes orchestrate tissue organisation.

## Homoiogenetic induction and lateral induction

**Homoiogenetic induction** (see also Box 1) is a process whereby '*committed or differentiated cells can cause one or few uncommitted cells in close proximity to differentiate like the majority*' (Gurdon et al, 1993). This is similar to the process that operates when slime moulds aggregate to form a fruiting body, where differentiating cells propagate pro-differentiation instructions in the form of secreted cAMP (Gerisch, 1986). Homoiogenetic induction differs somewhat from a classical community effect in that it is initiated by cells that have already started to differentiate, and is therefore a one-way instruction from differentiating cells to uncommitted cells, rather than a mutual decision-making process between large groups of uncommitted cells. However, in practice, these strategies may be difficult to distinguish experimentally.

Homoiogenetic induction may be a special case of a more general mechanism termed **lateral induction** (see Box 1), in which differentiating cells secrete short-range ligands that encourage their neighbours to differentiate in the same direction as themselves (Fig. 2G). If neighbours are also encouraged to upregulate expression of these ligands, then a differentiation response can be propagated

across a field of cells (Dieterle et al, 2020). For example, emerging prosensory cells of the inner ear upregulate the Notch ligand Jagged1: this signals to neighbouring cells to induce a prosensory fate while also inducing upregulation of Jagged1. This process results in the emergence of a coherent patch of prosensory cells within the non-sensory epithelium (Petrovic et al, 2014) (Fig. 2H,I). Examples of such 'differentiation-propagation' mechanisms can be found throughout and beyond development, including during differentiation of pluripotent cells (Liu et al, 2022; Schröter et al, 2023; Heemskerk, 2020), in the neural tube (Lehr et al, 2024), during formation of omatidia (Lubensky et al, 2011) and in the hair follicle (Deschene et al, 2014).

## Technical challenges in studying the synchronisation of differentiation

All of the strategies outlined above require cells to sense and respond to differentiation events in their neighbours. When, where, and how do cells do this? What is the molecular basis of the 'sensing' mechanisms? These questions have been difficult to answer due to technical challenges in monitoring the behaviour of cells in relation to their neighbours. In the next section, I discuss emerging approaches that overcome some of these technical challenges.

### Image analysis and single-cell transcriptomics

Perhaps the simplest approach to examining synchronisation is direct observation by imaging. Do differentiated cells emerge at the same time across a particular region (Fig. 3A), or do some differentiated cells emerge before others (Fig. 3B)? A number of image analysis methods have been designed to examine the properties of cells in relation to their neighbours (Forsyth et al, 2021; Gómez et al, 2021; Strauss et al, 2022; Summers et al, 2022; French et al, 2025): reviewed in (Malaguti et al, 2024). How can we visualise and quantify these neighbour relationships in 3D tissues (Fig. 3C,D)? A recently developed image analysis pipeline maps regions of local coherence or local heterogeneity. This approach uses the 'PRINGLE' algorithm (Projection and Relative Normalisation to aliGn muLtiple Epiblasts) to represent complex 3D curved regions in 2D space and to display averaged maps from multiple embryos or tissues, even where these differ in shape and size (Fig. 3E). This quantitative 'mapping' approach also makes it possible the measure when, where, and how local heterogeneity of differentiation responds to manipulating candidate regulators of cell-cell coordination (Fig. 3F) (French et al, 2025).

As an alternative approach, scRNAseq analysis of chimeric embryos at closely spaced time points can reveal when cells adjust the pace of differentiation to synchronise with neighbours, and can identify candidate regulators of this process. (Cheng et al, 2022). This approach is likely to become even more powerful with the development of single-cell resolution spatial transcriptomics (Rubinstein et al, 2025).

### Synthetic neighbour-labelling

One limitation to many imaging-based or transcriptome-based studies is that they are restricted to determining cell identity based on gene or protein expression, and can't be readily applied to functional assays of cell potency, commitment, or differentiation-

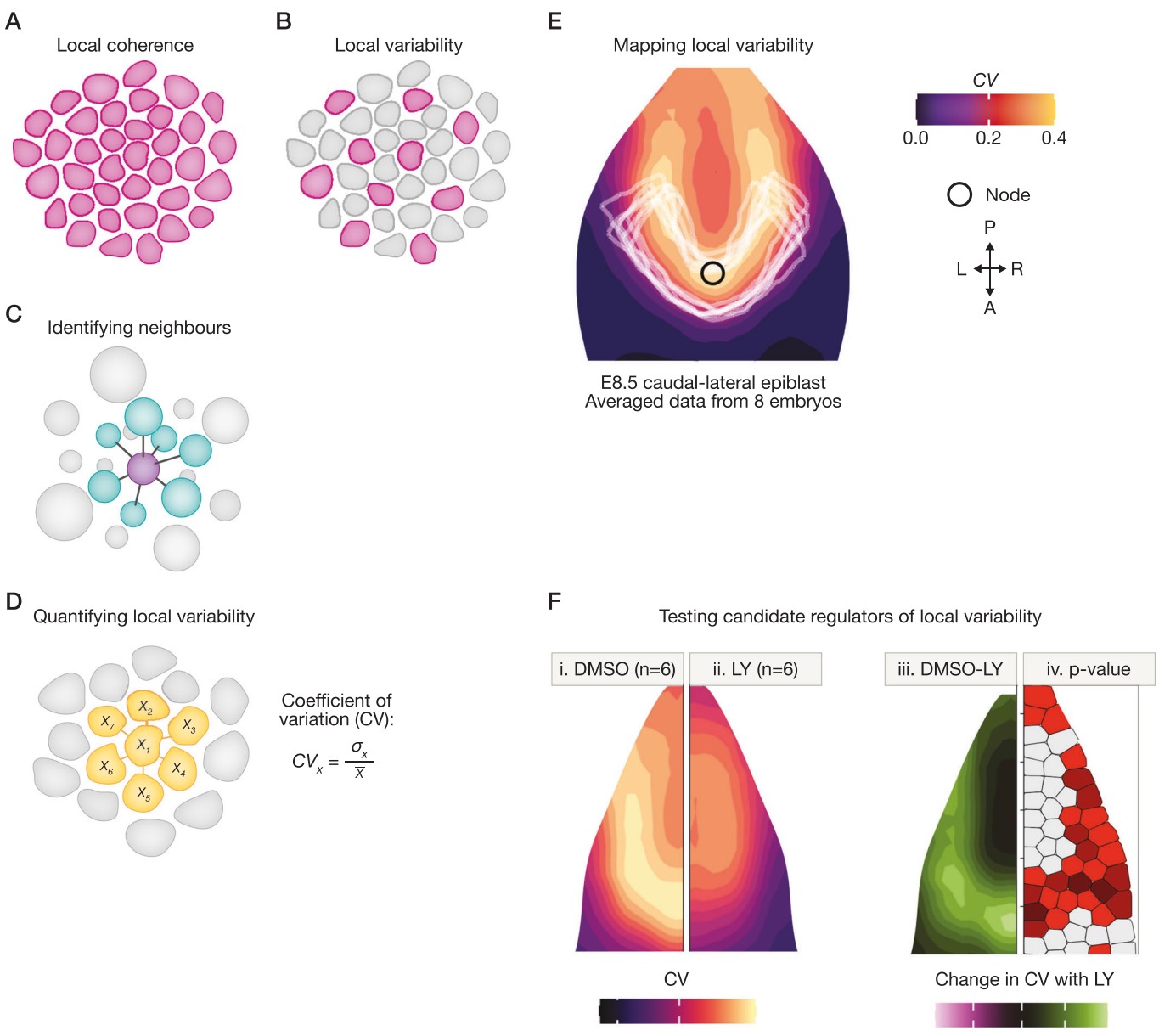

**Figure 3. Image analysis: to distinguish variable versus coherent differentiation.**

(A, B): Image analysis can be used to distinguish regions of coherent differentiation (A) from regions of variable or asynchronous differentiation (B). (C) Establishing neighbour relationships requires segmentation of individual cells and identification of neighbours. (D) Quantification of local variability. Coefficient for variability (CV) is used to compare the fluorescence intensity in each cell with the average fluorescence intensity in its immediate neighbours, providing a measure of local variability of gene or protein expression around each individual cell. (E) CV values can be mapped back onto tissues in order to map regions of coherence (dark) and regions of local variability (light orange). This image represents variability in Tbx6 expression, averaged from the caudal epiblast of 8 E8.5 mouse embryos (French et al, 2025). This shows regions of high local variability flanking the posterior midline. The PRINGLE algorithm makes it possible to superimpose averaged patterns across multiple embryos, normalising for differences in shape and size. Figure adapted from (French et al, 2025). (F) The approach shown in F makes it possible to compare local variability between experimental conditions or between different species. This image represents differences in variability of Tbx6 expression between six control embryos (i) and six embryos treated with Notch inhibitor LY (ii). iii maps regional differences between control and LY-treated inhibitors, and iv maps *p* values for these differences. Figure adapted from (French et al, 2025).

speed in live cells. The advent of 'neighbour labelling' technologies (reviewed in Malaguti et al, 2024) overcomes this limitation by preserving spatial information (i.e. who is a neighbour of whom) even after disaggregation of a tissue into individual living cells. This opens up opportunities for asking how differentiation of one cell influences the functional potency, as well as the transcriptional

state, of surrounding cells without being restricted to analysis of intact tissues (Fig. 4A).

Neighbour-labelling approaches are broadly based on strategies to induce or transmit a fluorescent label to the neighbours of a cell that is undergoing a particular change, such as a cell fate transition (Fig. 4A). One such approach is to engineer cells with synthetic

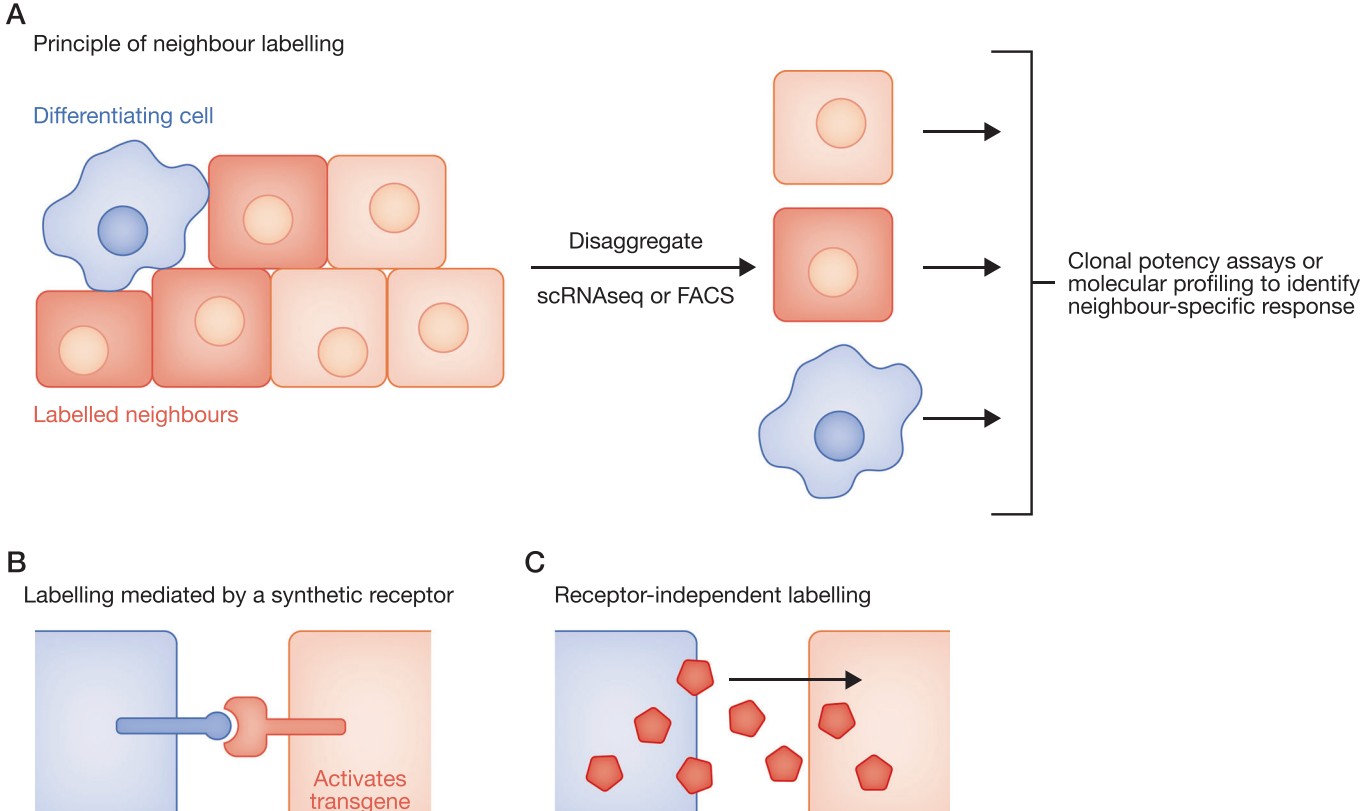

**Figure 4. Neighbour labelling.**

(A) Principle of neighbour labelling. A 'sender cell' (e.g. a differentiating cell: blue) causes surrounding nearby cells to become fluorescently labelled (red) while more distant cells remain unlabelled (pale pink). The tissue or culture of cells can then be disaggregated into single cells, preserving information about which cells were neighbours of the blue sender cells. Functional assays or molecular profiling can then be used to identify differences in potency or state between neighbours and non-neighbours to discover if and how differentiating cells influence differentiation in surrounding cells. Figure adapted from (Malaguti et al, 2024). (B) One approach to neighbour-labelling is based on synthetic receptors that upregulate a transgene upon detection of a synthetic or natural ligand on adjacent cells. If the transgene encodes a fluorescent protein, this results in contact-dependent fluorescent neighbour labelling. See text for examples. (C) An alternative approach to neighbour-labelling is to directly transfer a fluorescent molecule into neighbouring cells in a receptor-independent manner. See text for examples.

ligands (in 'sender cells') and synthetic receptors (in 'receivers' cells) that are coupled to a transgene that becomes activated when sender cells contact receiver cells (Fig. 4B) (reviewed in Malaguti et al, 2024; Trentesaux et al, 2023)). For example, the **SynNotch** system (Morsut et al, 2016) (see Box 1) is based on re-engineering the Notch receptor to detect a synthetic ligand (for example based on a GFP nanobody that detects extracellular GFP). Upon binding ligand, the SynNotch intracellular domain becomes cleaved and moves to the nucleus, where it activates a transgene (e.g. based on fusion to a transactivator). SyNotch-based systems have been adapted for use in pluripotent cells (Malaguti et al, 2022) and embryos (He et al, 2017; Zhang et al, 2022) and are therefore well-positioned for studying coordination of developmental transitions. This is one example of several elegant synthetic-signalling systems that could be used for studying embryonic development (Feinberg et al, 2008; Barnea et al, 2008; Daringer et al, 2014; Pasqual et al, 2018; Cachero et al, 2020; Tang et al, 2020; Minegishi et al, 2023) as reviewed in detail in (Malaguti et al, 2024).

Alternatively, simpler systems are based on receptor-independent transfer of a fluorescent protein between cells, using a second colour to distinguish senders from receivers (Fig. 4C)

(Ombrato et al, 2021; Lebek et al, 2024). The Cherry Niche system (Ombrato et al, 2021) delivers a liposoluble TATk-mCherry fusion from large masses of cells to their local niche, and is therefore well suited to studying tumour microenvironments(Ombrato et al, 2019; Rodrigues et al, 2024). **PUFFFIN** (positive ultrabright fluorescent fusion for identifying neighbours—see Box 1) relies on the negative charge of all cell membranes to enable universal receptor-independent, and uses HaloTag technology (Los et al, 2008) for flexible and highly sensitive colour-of-choice labelling. PUFFFIN is sensitive enough to detect the neighbours even of individual cells, and all PUFFFIN components are deliverable from a single plasmid, which can be delivered in vivo via electroporation or viral transduction (Lebek et al, 2024). These features of PUFFFIN make it particularly useful for studying local coordination of differentiation in embryos or cell-based models of development.

Neighbour-labelling technologies offer great promise for exploring the mechanistic basis of synchronisation strategies because they make it possible to profile any functional or molecular property of cells in relation to changes in neighbours (reviewed in (Malaguti et al, 2024)). These new approaches complement more traditional grafting approaches for studying how cells adjust to a change in

their local neighbourhood (Couly et al, 1998; Kinder et al, 2000; Cambray and Wilson, 2007; Voiculescu et al, 2014; Busby et al, 2024).

## Molecular mechanisms for synchronising differentiation

What types of molecular mechanisms can cells use to communicate their differentiation status to their neighbours? Many of the examples described above use mechanisms based on receptor-mediated cell signalling, in which cells use short-range diffusible signals or membrane-bound ligands to share information with their local neighbourhood (Schröter et al, 2023). Cell-to-cell propagation mechanisms can then transmit this information across a wider field of cells. For example, propagation of Erk activity generates coordinated waves of signalling that can orchestrate differentiation in the skin and bone (Gattiglio et al, 2023; Hiratsuka et al, 2015; Simone et al, 2021) (see more examples above in the section on lateral induction). Similarly, two-way positive feedback mechanisms enable mutual amplification of signals (Bolouri and Davidson, 2010; Saka et al, 2011; Hatakeyama et al, 2023) (Fig. 2C).

Changes in adhesion molecules can also influence signalling and consequently help coordinate differentiation (Takehara et al, 2015; Schäfer et al, 2014; Miroshnikova et al, 2018). For example the switch from E-cadherin to N-cadherin helps synchronise commitment to a neuroectoderm fate (Punovuori et al, 2019, 2021) and drives cell-cell propagation of mesodermal commitment (Martyn et al, 2019) during differentiation of pluripotent cells.

Mechanical forces transmit information across fields of cells, and can help synchronise cellular behaviours in a number of contexts (Uyttewaal et al, 2012; Fernández-Sánchez et al, 2015; Kim et al, 2018; Zamir et al, 2022). For example, mechanical constraints in the mammalian preimplantation embryo enforce timely and coherent changes in morphology and cell fate (Fabrèges et al, 2024), and constrained proliferation generates compressive forces that locally-coordinate specification of cell identity during tooth development (Shroff et al, 2024). Mechanical feedbacks also coordinate morphogenetic changes across fields of cells within 3D cell-based models such as intestinal organoids (Xue et al, 2025).

Cells also communicate via direct transfer of materials through gap junctions (Levin, 2007). For example, Drosophila blood progenitors use gap junctions to coordinate blood progenitor fate decisions via calcium signalling (Ho et al, 2021, 2023), and vertebrate neural stem cells gap junctions coordinate reactivation of neural stem cells in response to nutritional stimuli (Spéder and Brand, 2014). Transfer of materials through cytoplasmic bridges (Chaigne et al, 2019) or extracellular vesicles (Minakawa et al, 2021) could also help to coordinate and synchronise differentiation.

## Conclusions and perspectives

The mechanisms by which cells sense and respond to differentiation events in their neighbours are central to understanding the remarkable robustness of embryonic development (Waddington, 1959), and the regeneration and repair of adult tissues. These mechanisms are likely co-opted to drive pathological behaviours

**Box 2  In need of answers**

- When and where do cells differentiate synchronously during development? Addressing this question requires careful single-cell analysis of how cells differentiate in relation to their neighbours in intact tissues. This remains technically challenging.

- What are the mechanisms by which cells sense the pace or direction of differentiation in their neighbours? Relatively few such mechanisms are known (one famous example being juxtacrine Notch signalling). There is therefore a need for unbiased discovery of novel sensing mechanisms.

- When and where is synchronisation important for robust development? Addressing this question depends on understanding the mechanisms that govern synchronisation, then disrupting them to assess the consequences of experimental desynchronisation.

- Can we engineer synchronous behaviours in order to improve our control over differentiation in cell-based models of development, homeostasis, or disease?

such as tumorogenesis (Fernández-Sánchez et al, 2015; Lowell, 2020; Alladin et al, 2020).

Understanding the cellular and molecular mechanisms of local synchronisation could inform strategies to improve 3D models of development, such as organoids or embryo models, and could even help design therapeutic approaches for improving tissue repair and regeneration. These approaches will benefit from emerging technologies for synthetic control over cell-cell signalling in developmental systems (Morsut et al, 2016; Davies, 2017; Zhou et al, 2018; Toda et al, 2018; Malaguti et al, 2022; Trentesaux et al, 2023; Garibyan et al, 2024; Malaguti et al, 2024).

A recent preprint (Isomura et al, 2024) provides a beautiful example of using synthetic signalling to understand and control cell-cell synchronisation. The authors focused on the synchronised oscillations of Hes7 that characterise the natural segmentation clock within presomitic mesoderm (described in more detail above). They engineered optogenetic control over a re-purposed version of the Notch signalling pathway to reconstitute oscillatory gene expression in presomitic mesoderm derived from mouse embryonic stem cells (Matsumiya et al, 2018). This system allowed the authors to explore which components of the Notch signalling pathway are responsible for the synchronisation of oscillations across cells. This impressive study exemplifies the power of synthetic signalling for exploring mechanisms of cell-cell synchronisation and for engineering tunable control over coupled oscillations.

A recurring theme throughout this review is the observation that synchronisation mechanisms are understudied and poorly understood due to technical challenges (see also Box 2). The advent of new approaches for monitoring and manipulating differentiation of cells in relation to their neighbours (Malaguti et al, 2024) seems likely to open up a new era of understanding how cells communicate to generate coherent tissues.

## Peer review information

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

## Acknowledgements

Thanks to Dr Jennifer Annoh and Tamina Lebek for help with figures and Val Wilson for helpful discussions. SL is funded by Wellcome Trust Senior Fellowship 220298 from the Wellcome Trust (https://wellcome.org/).

## Author contributions

**Sally Lowell**: Conceptualisation; Funding acquisition; Writing—original draft; Writing—review and editing.

## Disclosure and competing interests statement

The authors declare no competing interests.

