## [Peer Review File · EMBO Reports]

Keeping up with the neighbours: Local synchronisation of cell fate decisions during development

Sally Lowell

Corresponding author(s): Sally Lowell (sally.lowell@ed.ac.uk)

Review Timeline:

Submission Date:	22nd Jun 25
Editorial Decision:	22nd Jul 25
Revision Received:	23rd Oct 25
Accepted:	21st Nov 25

Editor: Achim Breiling

Transaction Report:

Dear Prof. Lowell,

Thank you for the submission of your review article to our editorial offices. I have now received the full set of referee reports that is copied below. As you will see, all three referees state that your manuscript is interesting and timely. However, they have several suggestions to improve the submission that I kindly ask you to address in a revised manuscript.

Given the constructive referee comments, I would thus like to invite you to revise your manuscript with the understanding that all referee points will be addressed in the revised manuscript and in a detailed point-by-point response.

I further have these editorial requests:

- Please add up to 5 keywords to the manuscript and place these below the abstract.
 - We have space for 1 more figure, and it would be nice to have indeed 4 figures, as we encourage authors to maximize the use of visual elements, which will increase the accessibility of the piece to a non-specialist readership. Please consider adding 1 more figure and note the instructions regarding figures below.
 - We usually ask our authors to include a box called "In need of answers" that briefly outlines the major questions that are still open in a given field in the form of a few bullet points. These questions can be accompanied by a brief explanation of what would be needed to address them and may provide helpful towards setting the stage for future experimentation in the field. For an example see this recent review we published: <https://www.embopress.org/doi/full/10.1038/s44319-024-00135-4>
 - Please also add callouts for the box to the manuscript text (Box 1).
 - Please also make sure the references and their callouts are formatted according to our reference format (with et al. for manuscripts with more than 10 authors):
<http://www.embopress.org/page/journal/14693178/authorguide#referencesformat>
 - We updated our journal's competing interests policy in January 2022 and request authors to consider both actual and perceived competing interests. Please review the policy <https://www.embopress.org/competing-interests> and update your competing interests if necessary. Please name this section 'Disclosure and Competing Interests Statement' and put it after the Acknowledgements section.
 - Please make sure that all the funding information is also entered into the online submission system and that it is complete and similar to the one in the acknowledgement section of the manuscript text file.
- I think this is a very interesting review and while I appreciate that incorporating the referees' suggestions will still require some work, I am convinced that the article is worth it and will benefit from it.

When submitting your revised manuscript, we will require a Microsoft Word file (.doc) of the revised manuscript text including detailed figure legends (at the very end), but without the figures.

Please provide the final figures as separate, high-resolution files as .pdf, .eps, .tif, or .jpg (one file per figure). Please finalize the drafts provided and make sure they accurately illustrate the key scientific concepts that you wish to show.

Please also note the following points:

- If there are certain aspects of your figure draft that are based upon assumptions or where the scientific data remains ambiguous (for example, schematically depicting a presumed direct protein-protein interaction, protein shape or subcellular localizations etc.) please add a comment so that we can work with you on an accurate depiction. Please ensure the directionality and nature of interactions is presented accurately.
- If the figure or single panels of the figure have been adapted from a published figure, please add this information to the figure legend (e.g., 'Adapted from...' or 'Based on...'). The editor will discuss if a reference and permission will be necessary
- Please only re-use figures or parts of a figure if this is essential for understanding the concept communicated. Often a reference to a previous paper will suffice. If the figure contains re-used images or elements of images, including schematics, micrographs or photos, please make sure that you have the permission/license to publish it (this also applies to your own previous work, if the journal you published in retains copyright. Certain 'creative commons' open access licenses, such as CC-BY 4.0, allow re-use without additional formal permissions). All re-used material must be explicitly cited.

- If you use an image data base for scientific iconography (e.g., BioRender), please let us know if you have a license that allows for publication in an academic journal. Often authors use misleading iconography for expedience. Please ensure the information shown is scientifically accurate. If in doubt, please discuss with the editor or provide a sketch so that our designers can create accurate iconography.

- For figures created using a software for editing vector objects like Inkscape, CorelDraw etc., please send the file as a PDF (or SVG, or EPS), PowerPoint or Keynote in which the labels and objects are still editable. For figures created using Adobe Illustrator, please send the Illustrator (.ai) file.

I look forward to seeing a revised version of your manuscript when it is ready. Please let me know if you have questions or comments regarding the revision.

Kind regards,

Achim Breiling

Referee #1:

This is an extremely thoughtful and interesting review on the mechanisms driving synchronization of differentiation. It covers a largely unexplored concept, but one that is potentially very important, as it would explain the uniformity in cell identities found in differentiating tissues. The review does an excellent job in covering not only what is known about the subject, but also in describing the tools that may allow to tackle this question in more depth in the future.

I only have one suggestion for a minor correction, in line 121 the authors may want to consider citing PMID: 28919206, as this manuscript describes a role for cell competition in eliminating cells that differentiate precociously.

Referee #2:

In this review, Sally Lowell provides a clear, well-organised, and enjoyable overview of a very critical and understudied mechanism of cell-to-cell communication: synchronised or coupled differentiation, revisiting concepts such as community effects, quorum sensing, oscillatory coupling, and lateral induction. This phenomenon has been observed since over 35 years, thanks to the pioneering observations of Prof. John Gurdon in engrafted *Xenopus* embryos. Importantly, it highlights the need to go beyond a cell-autonomous view of differentiation and consider instead the emergent properties of higher order structures.

Several technical challenges have hampered the swift investigation of local synchronisation or coordination of cell fate choices. I was expecting, given the important contribution of the author herself to the field, that this review would highlight the recent technological advances and the new original approaches that allow us now to explore local differentiation and signal oscillations in several tissues. I was therefore slightly disappointed to realise that the original label-neighbouring technologies, such as SynNotch, Puffin or even BacTrace, are only marginally mentioned in a figure legend and not thoroughly described in the review. Strategies based on measuring single cell properties in relation to the properties of their neighbours to quantify tissue patterning should be more extensively presented. Similarly, the molecular mechanisms underlying the capacity of cells to sense and respond to the state of differentiation of their neighbours are only superficially reviewed, with little or no concrete examples.

A few examples of important perspectives that could be developed further, are listed below:

1. One under-explored angle is how the mechanisms of synchronisation impact the resulting spatial patterns of tissues: while some mechanisms produce salt-and-pepper arrangements, others generate tissue borders or morphogen gradients. Adding a classification or comparison between distinct tissue patterns and the mechanisms driving their formation could provide a richer conceptual vision.
2. Some transitions between cell-level mechanisms and tissue-level outcomes come about abruptly. Specifically, the outputs of the quorum sensing examples are not clearly explained. Are the cells forming a pattern, are they all differentiating toward the same fate? Clarifying what operates at the single cell scale vs. tissue scale and whether different mechanisms dominate at each scale would help the reader better navigate the complexity of the topic. For example, in the section about hair follicles

regeneration or the following paragraph on immune cells, better precision would be appreciated.

3. The origins of synchronisation remain under-discussed. Is there always a signalling centre from which signals originate? Is cell neighbour's behaviour truly emergent or triggered by local organisers? The tooth placode is a striking example where a collective wave of differentiation is coordinated by a signalling hub: this has been elegantly shown by the Ophir Klein and Otger Campàs labs (<https://doi.org/10.1038/s41556-024-01380-4>). Similar tissue-wide coordination of fate commitment has been described in the work from the Liberali lab (<https://doi.org/10.1038/s41567-025-02792-1>; <https://doi.org/10.1101/2025.01.14.632683>) and the Fre lab (<https://doi.org/10.1101/2024.07.21.603898>) and could be mentioned to complete the picture.

4. Finally, in the section on oscillations, a short paragraph introducing recent work by Ina Sonnen lab (e.g. tissue-scale oscillatory coordination in intestinal organoids) would nicely enrich the discussion of coupled oscillators beyond the classical PSM examples (<https://doi.org/10.1101/2024.08.26.609553>).

Regarding Part 1: The need to synchronise differentiation

This section should be slightly reorganised: I would recommend beginning with a clear definition of local synchronisation during differentiation, why it is important and when it is observed. Examples could be provided there, to build a first description of synchronisation, followed by the explanation of why synchronisation is challenged, when it is critical to maintain it and how.

I provide a few more specific comments below:

- The paragraph at lines 59-64 is quite vague. For instance, the statement "progenitor cells need to generate more than one cell type in the same region" should be elaborated in the context of organ development to support tissue function and ideally include some concrete examples.
- The terms lateral inhibition and Turing patterning should be briefly explained to ensure clarity for all readers.
- The sentence "Variability is also essential to enable symmetry breaking" is unclear. How and why is variability essential in this context? An explanation is needed.
- The phrase at lines 66-68 is also elusive: please provide specific examples.
- At lines 94-99, it is unclear how cells sense their neighbours? The findings from Malaguti et al. could be summarised in one sentence here to improve intelligibility.

Regarding Part 2: Strategies for synchronising differentiation

- Lines 173-174 repeat the exact same sentence from lines 134-135. Additionally, a paragraph at the end of the quorum sensing section already draws a parallel between the community effects and quorum sensing, making this repetition unnecessary.
- Line 217: the sentence "They communicate via FGF4 to control proliferation" should be elaborated further. How does this communication occur? In which way is proliferation controlled via FGF4? In general, examples from published studies should be illustrated in more detail for clarity.
- The sentence at lines 232-234 should be strengthened by one or two references.
- At line 237 I recommend adding one or two more recent references, besides Isomura et al., such as Boseman and Sonnen (2022) and Chandel et al. (Development, 2024) or the recent biorxiv pre-print from the Sonnen lab cited above: <https://doi.org/10.1101/2024.08.26.609553>.
- At line 239-245: the publication by Sonnen et al, Cell 2018 should be cited.

In Part 3: Technical challenges in studying synchronisation of differentiation

In the synthetic neighbour-labelling section, it would be beneficial to add a paragraph on synthetic technologies developed to study cell-cell interactions in vivo. In this context, the following studies should be cited: Zhang et al., Science, 2018 and Huang et al., PNAS, 2017 for the SynNotch mice and Pasqual et al., Nature, 2017 for the LIPSTIC method.

Overall, I found this review to address a highly important and interesting aspect of cell fate determination. However, it comes across as somewhat superficial, as if written in a hurry. Even complex concepts such as quorum sensing and homoiogenetic induction are mentioned without sufficient elaboration.

Related to this, the text needs careful proof reading, to correct the numerous typos, of which I am listing some examples below:

- Line 35: Part 1 is numbered 1 but the following Parts are not numbered
- Line 79: that it is formed in in

- Line 81: define the acronym GRN
- Line 92: Mulas et al, no date
- Line 101: "also" is written twice
- Line 128: comma at the wrong place, after for
- Line 202: "Decisive decisions" is redundant
- Line 228: coordination instead of coordinating
- Line 258: "cell-cell coupling of is critical". Delete "of".
- Line 326: "When and where and how to cells do this?" please rephrase.
- Line 428: "using synthetic signalling understand". "to" is missing.

Referee #3:

Dr. Lowell describes the concept and importance of synchronization in cell differentiation during development. The manuscript provides a good summary of scenarios in which developing tissues require synchronization, as well as the strategies multicellular systems employ to achieve it.

However, the descriptions of individual mechanisms remain abstract. In particular, the current figures illustrate the overall concept of synchronization and the strategies involved, but they are too simple and have no information about what tissues exhibit synchronized differentiation or what molecular mechanisms underlie such synchronization. The molecular mechanisms are only explained in the text and may be difficult to follow for non-expert readers. I suggest revising Figure 2 to provide more detail on key mechanisms of synchronization, such as mutual positive feedback, cell-cell coupling of molecular oscillators in the segmentation clock, and homoiogetic induction.

Additionally, regarding Figure 3, there is currently no explanation of how the synNotch and PUFFIN systems work. My suggestion is adding more detailed descriptions of these systems, including their system design and the types of cell-cell communication they visualize. Figure 3B shows a difference in labeled cells between the synNotch and PUFFIN systems, but additional explanation is required for this comparison. It would also be helpful to highlight the respective strengths and limitations of each system.

As for the references, I recommend citing original research articles, especially for key findings, although I understand the space constraints in some cases. For example, in Line 120, three review articles are cited regarding the elimination of misdifferentiating cells by cell competition. Are there original research articles that demonstrate cell competition during development that could be cited? Similarly, in Line 355, Malaguti et al. (2024), a review article, is cited for neighbor labeling technologies. However, original research articles for the development of specific technologies-such as split GFP reconstitution, synNotch, PAGER, MESA, PUFFIN, and others-should be referenced instead. Regarding the citation of Fulton et al. (2021) and Lowell & Blin (2022) in Line 47, is Fulton et al. (2021) still a preprint, or has it been published in a peer-reviewed journal? If not, is it possible to cite alternative, recently published studies?

I also noticed several typos in the manuscript:

- Line 79, "in" is repeated.
- Line 101, "also" is repeated.
- Line 258, "of" should be removed.
- Line 403, "also" is repeated.

Please carefully check the manuscript for remaining typos.

EMBOR-2025-62177 Lowell. Response to reviewers and editor

Author responses are in blue italics

Response to requests from editor

- Please add up to 5 keywords to the manuscript and place these below the abstract.

Done

- We have space for 1 more figure, and it would be nice to have indeed 4 figures, as we encourage authors to maximize the use of visual elements, which will increase the accessibility of the piece to a non-specialist readership. Please consider adding 1 more figure

I have added a new figure and expanded the content of all figures.

- We usually ask our authors to include a box called "In need of answers" that briefly outlines the major questions that are still open in a given field in the form of a few bullet points. These questions can be accompanied by a brief explanation of what would be needed to address them and may provide helpful towards setting the stage for future experimentation in the field.

Done

- Please also add callouts for the box to the manuscript text (Box 1).

Done

- Please also make sure the references and their callouts are formatted according to our reference format (with et al. for manuscripts with more than 10 authors):

I have formatted references according to the "EMBO" setting in my reference manager. If additional manual editing of the reference style is required it would be best for me to do this after the manuscript is complete and no more reference changes are needed.

- We updated our journal's competing interests policy in January 2022 and request authors to consider both actual and perceived competing interests. Please review the policy <https://www.embopress.org/competing-interests> and update your competing interests if necessary. Please name this section 'Disclosure and Competing Interests Statement' and put it after the Acknowledgements section.

Done

- Please make sure that all the funding information is also entered into the online submission system and that it is complete and similar to the one in the acknowledgement section of the manuscript text file.

Done

Response to Referee #1:

This is an extremely thoughtful and interesting review on the mechanisms driving synchronization of differentiation. It covers a largely unexplored concept, but one that is

potentially very important, as it would explain the uniformity in cell identities found in differentiating tissues. The review does an excellent job in covering not only what is known about the subject, but also in describing the tools that may allow to tackle this question in more depth in the future.

I only have one suggestion for a minor correction, in line 121 the authors may want to consider citing PMID: 28919206, as this manuscript describes a role for cell competition in eliminating cells that differentiate precociously.

Thank you. The suggestion citation has now been included.

Response to Referee #2:

In this review, Sally Lowell provides a clear, well-organised, and enjoyable overview of a very critical and understudied mechanism of cell-to-cell communication: synchronised or coupled differentiation, revisiting concepts such as community effects, quorum sensing, oscillatory coupling, and lateral induction. This phenomenon has been observed since over 35 years, thanks to the pioneering observations of Prof. John Gurdon in engrafted *Xenopus* embryos. Importantly, it highlights the need to go beyond a cell-autonomous view of differentiation and consider instead the emergent properties of higher order structures. Several technical challenges have hampered the swift investigation of local synchronisation or coordination of cell fate choices. I was expecting, given the important contribution of the author herself to the field, that this review would highlight the recent technological advances and the new original approaches that allow us now to explore local differentiation and signal oscillations in several tissues. I was therefore slightly disappointed to realise that the original label-neighbouring technologies, such as SynNotch, Puffin or even BacTrace, are only marginally mentioned in a figure legend and not thoroughly described in the review.

Strategies based on measuring single cell properties in relation to the properties of their neighbours to quantify tissue patterning should be more extensively presented. Similarly, the molecular mechanisms underlying the capacity of cells to sense and respond to the state of differentiation of their neighbours are only superficially reviewed, with little or no concrete examples.

A few examples of important perspectives that could be developed further, are listed below:

1. One under-explored angle is how the mechanisms of synchronisation impact the resulting spatial patterns of tissues: while some mechanisms produce salt-and-pepper arrangements, others generate tissue borders or morphogen gradients. Adding a classification or comparison between distinct tissue patterns and the mechanisms driving their formation could provide a richer conceptual vision.

Thank you for this suggestion. This chimes with my own thoughts when originally writing this article, when I envisaged a broader scope encompassing different aspects of patterning, as suggested by the reviewer. However, I made the decision to focus primarily on "local synchronisation" rather than a broader overview of different patterning mechanisms. This is explained in lines 74-93 which briefly outline other patterning systems with citations to relevant reviews. I take the reviewers point that this could have been explained more clearly, so I've added more text at lines 94-95 to clarify the particular focus of this article.

I've also addressed this point by including a glossary that briefly outlines different categories of patterning mechanisms, in addition to lines 74 - 95, which I hope provides a glimpse of how the focus of the article fits within the broader context.

2. Some transitions between cell-level mechanisms and tissue-level outcomes come about abruptly. Specifically, the outputs of the quorum sensing examples are not clearly explained. Are the cells forming a pattern, are they all differentiating toward the same fate?

I have added a sentence (lines 260-61) to explain that this article is predominantly discussing mechanisms that enable cells to differentiate towards the same fate at the same time.

Clarifying what operates at the single cell scale vs. tissue scale and whether different mechanisms dominate at each scale would help the reader better navigate the complexity of the topic. For example, in the section about hair follicles regeneration or the following paragraph on immune cells, better precision would be appreciated.

I have expanded the discussion about hair follicles with more mechanistic detail (lines 224-227). The work on immune cells is based mainly on theoretical considerations supported by mathematical modelling, so I feel the conceptual level of our explanation is appropriate: explaining the mathematics in more detail would be overly-complicated for this review. However I have slightly reworded the description to improve clarity, and added a perspective on relevance to developmental decisions (lines 234-237)

3. The origins of synchronisation remain under-discussed. Is there always a signalling centre from which signals originate? Is cell neighbour's behaviour truly emergent or triggered by local organisers? The tooth placode is a striking example where a collective wave of differentiation is coordinated by a signalling hub: this has been elegantly shown by the Ophir Klein and Otger Campàs labs (<https://doi.org/10.1038/s41556-024-01380-4>

This is an interesting point that I agree could have been brought out more clearly in the original article. I've added in lines 86-92 that discuss this general point. I also now refer to the Klein and Campàs reference within our revised manuscript (lines 466-8).

Similar tissue-wide coordination of fate commitment has been described in the work from the Liberali lab (<https://doi.org/10.1038/s41567-025-02792-1>; <https://doi.org/10.1101/2025.01.14.632683>) and the Fre lab (<https://doi.org/10.1101/2024.07.21.603898>) and could be mentioned to complete the picture.

Thank you for these suggestions, which have all been incorporated into the revised manuscript. The paper by Xue et al discusses coordination of morphogenesis (rather than differentiation), and I agree that this adds an interesting perspective that I missed in our original article. I have included this point, and reference (lines 468-9). The two papers by Schwyer et al and Journot et al discuss emergence of variability, rather than synchronisation, so I have added these to the early section of the manuscript where I briefly introduce the idea of emergent variability as a contrast to synchronisation (lines 80-82).

Finally, in the section on oscillations, a short paragraph introducing recent work by Ina Sonnen lab (e.g. tissue-scale oscillatory coordination in intestinal organoids) would nicely enrich the discussion of coupled oscillators beyond the classical PSM examples (<https://doi.org/10.1101/2024.08.26.609553>).

I have followed this helpful suggestion and have extended this section to include mention of other tissues containing oscillators, including intestinal organoids, muscle stem cells, and pancreas progenitors (lines 330-34).

Regarding Part 1: The need to synchronise differentiation

This section should be slightly reorganised: I would recommend beginning with a clear definition of local synchronisation during differentiation, why it is important and when it is observed. Examples could be provided there, to build a first description of synchronisation, followed by the explanation of why synchronisation is challenged, when it is critical to maintain it and how.

This was a very useful suggestion. I agree a definition of synchronisation is important: I have now included this in our new "glossary" section, and refer to it at the start of the article.

*Turning to the question of *when* synchronisation is observed: this brings up a point that was not clearly articulated in the original manuscript, which is that it is actually rather difficult to know when cells are locally synchronising. I have now directly addressed this point in the section "In need of answers" (lines 542-544). I begin the article with one example where I believe the case for synchronisation is particularly clear (the peri-implantation epiblast), and turn to other examples throughout the article.*

I provide a few more specific comments below:

- The paragraph at lines 59-64 is quite vague. For instance, the statement "progenitor cells need to generate more than one cell type in the same region" should be elaborated in the context of organ development to support tissue function and ideally include some concrete examples.

I have rewritten this section and hope that it is now clearer (lines 74-82). I aim to keep this section as brief as possible because the generation of diversity is not the topic of this review, and is only mentioned to provide context (i.e. to point out that synchronisation isn't always beneficial). For this reason, I chose not to further expand this section to discuss particular examples, but instead refer the reader to reviews where examples can be found.

- The terms lateral inhibition and Turing patterning should be briefly explained to ensure clarity for all readers.

I have briefly defined these terms in the new "Glossary" section. I have chosen to convey the role of these processes rather than describing the mechanisms in detail because these mechanisms relate to generation of diversity, which is not the main topic of the article. I instead refer the reader to references where more details can be found.

- The sentence "Variability is also essential to enable symmetry breaking" is unclear. How and why is variability essential in this context? An explanation is needed.

I have rewritten this section (lines 74-82)

- The phrase at lines 66-68 is also elusive: please provide specific examples.

The original manuscript did not make it clear that these lines were intended to serve as an introduction to the examples that would be discussed throughout the rest of the review. I have expanded on this point in our revised manuscript (lines 330-334).

- At lines 94-99, it is unclear how cells sense their neighbours? The findings from Malaguti et al. could be summarised in one sentence here to improve intelligibility.

These findings have now been summarised with an additional sentence (line 124-127), thank you for this suggestion.

Regarding Part 2: Strategies for synchronising differentiation

- Lines 173-174 repeat the exact same sentence from lines 134-135.

Additionally, a paragraph at the end of the quorum sensing section already draws a parallel between the community effects and quorum sensing, making this repetition unnecessary.

Thank you for spotting this unnecessary repetition-I have deleted lines 173-174.

- Line 217: the sentence "They communicate via FGF4 to control proliferation" should be elaborated further. How does this communication occur? In which way is proliferation controlled via FGF4? In general, examples from published studies should be illustrated in more detail for clarity.

I have expanded on this point and discussed it in the context of other published papers (lines 249-258).

- The sentence at lines 232-234 should be strengthened by one or two references.

This sentence is now supported by two references (line 274).

- At line 237 I recommend adding one or two more recent references, besides Isomura et al., such as Boseman and Sonnen (2022) and Chandel et al. (Development, 2024) or the recent biorxiv pre-print from the Sonnen lab cited above:
<https://doi.org/10.1101/2024.08.26.609553>.

Thank you, I have added two additional references (Bosman & Sonnen, and Chandel et al), both excellent reviews (line 278).

- At line 239-245: the publication by Sonnen et al, Cell 2018 should be cited.

This citation has been added

In Part 3: Technical challenges in studying synchronisation of differentiation
In the synthetic neighbour-labelling section, it would be beneficial to add a paragraph on synthetic technologies developed to study cell-cell interactions in vivo. In this context, the following studies should be cited: Zhang et al., Science, 2018 and Huang et al., PNAS, 2017 for the SynNotch mice and Pasqual et al., Nature, 2017 for the LIPSTIC method.

Thank you for this suggestion. I have added a paragraph on this topic (Lines 407-434) and referenced several additional primary papers, including those suggested by this reviewer. We have recently written a comprehensive review on this exact topic (Malaguti et al 2024) so I kept this overview brief within the current manuscript.

Overall, I found this review to address a highly important and interesting aspect of cell fate determination. However, it comes across as somewhat superficial, as if written in a hurry. Even complex concepts such as quorum sensing and homoiogenetic induction are mentioned without sufficient elaboration.

I have now defined these terms in the new glossary.

Related to this, the text needs careful proof reading, to correct the numerous typos, of which I am listing some examples below:

- - Line 35: Part 1 is numbered 1 but the following Parts are not numbered
- - Line 79: that it is formed in in
- - Line 81: define the acronym GRN
- - Line 92: Mulas et al, no date
- - Line 101: "also" is written twice
- - Line 128: comma at the wrong place, after for

- - Line 202: "Decisive decisions" is redundant
- - Line 228: coordination instead of coordinating
- - Line 258: "cell-cell coupling of is critical". Delete "of".
- - Line 326: "When and where and how to cells do this?" please rephrase.
- - Line 428: "using synthetic signalling understand". "to" is missing.

Many thanks, I have corrected the above errors and proof-read the revised article for additional errors.

Response to Referee #3:

Dr. Lowell describes the concept and importance of synchronization in cell differentiation during development. The manuscript provides a good summary of scenarios in which developing tissues require synchronization, as well as the strategies multicellular systems employ to achieve it. However, the descriptions of individual mechanisms remain abstract. In particular, the current figures illustrate the overall concept of synchronization and the strategies involved, but they are too simple and have no information about what tissues exhibit synchronized differentiation or what molecular mechanisms underlie such synchronization. The molecular mechanisms are only explained in the text and may be difficult to follow for non-expert readers. I suggest revising Figure 2 to provide more detail on key mechanisms of synchronization, such as mutual positive feedback, cell-cell coupling of molecular oscillators in the segmentation clock, and homoiogenetic induction.

Thank you for this suggestion. I have expanded Figure 2 to provide examples of the three types of mechanisms plus information about molecular mechanisms that underlie them.

Additionally, regarding Figure 3, there is currently no explanation of how the synNotch and PUFFIN systems work. My suggestion is adding more detailed descriptions of these systems, including their system design and the types of cell-cell communication they visualize. Figure 3B shows a difference in labeled cells between the synNotch and PUFFIN systems, but additional explanation is required for this comparison. It would also be helpful to highlight the respective strengths and limitations of each system.

Thank you for this suggestion. I have expanded on this topic within the text and the figures (lines 407-434). We have recently written a comprehensive review on this exact topic (Malaguti et al 2024) so I kept this overview relatively brief within the current manuscript.

As for the references, I recommend citing original research articles, especially for key findings, although I understand the space constraints in some cases. For example, in Line 120, three review articles are cited regarding the elimination of misdifferentiating cells by cell competition. Are there original research articles that demonstrate cell competition during development that could be cited?

This was also suggested by reviewer 1: I have modified the reference list accordingly. Due to space limitations I have added one original research article (Diaz-Diaz et al) that focuses directly on the elimination of preciously differentiating cells (directly relevant to the topic of this review). There are of course several other important publications on other aspects of cell competition during development: for the sake of space I cover these by citing reviews.

Similarly, in Line 355, Malaguti et al. (2024), a review article, is cited for neighbor labeling technologies. However, original research articles for the development of specific technologies—such as split GFP reconstitution, synNotch, PAGER, MESA, PUFFIN, and others—should be referenced instead.

I have expanded this section (lines 407-434) and referenced several additional primary papers.

Regarding the citation of Fulton et al. (2021) and Lowell & Blin (2022) in Line 47, is Fulton et al. (2021) still a preprint, or has it been published in a peer-reviewed journal? If not, is it possible to cite alternative, recently published studies?

This is still a preprint and has not yet been published in a peer-reviewed journal. There are no published studies that recapitulate this observation. I have kept this citation within the article.

I also noticed several typos in the manuscript:

- Line 79, "in" is repeated.
- Line 101 "also" is repeated.
- Line 258, "of" should be removed.
- Line 403 "also" is repeated.
- Please carefully check the manuscript for remaining typos.

Thank you. I have corrected these and several other typos.

Prof. Sally Lowell
University of Edinburgh
Institute for Stem Cell Research, Centre for Regenerative Medicine, Institute for Regeneration and Repair
5 Little France Drive
Edinburgh EH16 4UU
United Kingdom

Dear Prof. Lowell,

Thank you for the submission of your revised manuscript to our editorial offices. I have now received the reports from the two referees that I asked to re-assess the article, you will find below. As you will see, the referees now fully support publication of your review in EMBO reports.

I am thus pleased to inform you that your review manuscript has been accepted for publication in EMBO reports. Your manuscript will be processed for publication by EMBO Press. It will be copy edited and you will receive page proofs prior to publication.

You will soon be contacted by Springer Nature to sign your publishing license. When you login to the customer service website, please use the token/code copied below to waive the article publication charges. Should you experience any difficulty, please email publishing@embo.org.

If you have any further questions, please do not hesitate to contact the Editorial Office. Thank you for your contribution to EMBO Reports.

Yours sincerely,

Referee #2:

The authors have greatly improved their manuscript following suggestions of three Referees who largely agreed on the corrections required. I particularly appreciated the new Glossary section. This review is now ready to be published in my opinion.

Referee #3:

I have reviewed the revised figures and the author's responses to the reviewers. The figures have been significantly improved to illustrate the examples of synchronized differentiation and the concepts of neighbor labeling. I believe the revised manuscript is suitable for publication.
